# Mitochondrial Haemoglobin Is Upregulated with Hypoxia in Skeletal Muscle and Has a Conserved Interaction with ATP Synthase and Inhibitory Factor 1

**DOI:** 10.3390/cells12060912

**Published:** 2023-03-16

**Authors:** Brad Ebanks, Gunjan Katyal, Chris Taylor, Adam Dowle, Chiara Papetti, Magnus Lucassen, Nicoleta Moisoi, Lisa Chakrabarti

**Affiliations:** 1School of Veterinary Medicine and Science, University of Nottingham, Nottingham NG7 2RD, UK; 2Department of Biology, Bioscience Technology Facility, University of York, York YO10 5DD, UK; 3Biology Department, University of Padova, 35122 Padova, Italy; 4Alfred Wegener Institute, 27568 Bremerhaven, Germany; 5Leicester School of Pharmacy, Leicester Institute for Pharmaceutical Innovation, De Montfort University, The Gateway, Leicester LE1 9BH, UK; 6MRC Versus Arthritis Centre for Musculoskeletal Ageing Research, Liverpool L7 8TX, UK

**Keywords:** mitochondria, ATP synthase, haemoglobin, hypoxia, ATP synthase inhibitory factor 1

## Abstract

The globin protein superfamily has diverse functions. Haemoglobin has been found in non-erythroid locations, including within the mitochondria. Using co-immunoprecipitation and in silico methods, we investigated the interaction of mitochondrial haemoglobin with ATP synthase and its associated proteins, including inhibitory factor 1 (IF1). We measured the expression of mitochondrial haemoglobin in response to hypoxia. In vitro and in silico evidence of interactions between mitochondrial haemoglobin and ATP synthase were found, and we report upregulated mitochondrial haemoglobin expression in response to hypoxia within skeletal muscle tissue. Our observations indicate that mitochondrial pH and ATP synthase activity are implicated in the mitochondrial haemoglobin response to hypoxia.

## 1. Introduction

Haemoglobin has a physiological role within red-blood cells as the protein that binds to both oxygen and carbon dioxide. This function of haemoglobin allows for the controlled delivery of oxygen to tissues to meet metabolic demand. Its structure was first characterised in the 1960s [1]; shortly after, it was demonstrated that its function could be modulated by pH and the allosteric binding of organic phosphates [2,3,4]. The tetrameric α_2_β_2_ structure, with a prosthetic haem group located within each subunit, allows for a co-operative oxygen-binding mechanism that results in oxyhaemoglobin having a substantially different structure to deoxyhaemoglobin [5].

In the last two decades, myoglobin and haemoglobin have been grouped with novel globin proteins such as cytoglobin [6], neuroglobin [7], and globin X [8]. While the precise physiological purpose of this expanded globin family is yet to be fully understood, it is suggested they may have signalling functions and the capacity to buffer against oxidative stress [9], while their dysfunction is implicated in disease pathologies [10]. Alongside an expanded family of globin proteins, there has also been an increased understanding of non-erythroid haemoglobin. Haemoglobin has been located in neuronal cells [11,12,13] and within the mitochondria [14]. Both neuronal haemoglobin and mitochondrial haemoglobin have been suggested to be dysfunctional in the pathophysiology of Parkinson’s disease [15,16,17,18]. Despite this, the intracellular physiological role of these haemoglobin proteins remains undefined.

Comparative physiology can be used to understand the functionality of non-erythroid haemoglobin. The function and expression of haemoglobin varies across species. *D. melanogaster* possess three globin genes: a haemoglobin, Glob1, which is widely expressed across the tracheal system and fat body of the *D. melanogaster* and is thought to play an important role in O_2_ homeostasis [19,20], while Glob2 and Glob3 show exclusive and limited expression in the testes of male *D. melanogaster* [21].

When using a comparative physiology approach to study haemoglobin, the Channichthyidae family of fishes from the Notothenioid suborder offer a unique advantage. This is because Channichthyidae fishes are unique among vertebrates in their absence of haemoglobin expression, and in some species, their myoglobin is at low levels or not expressed [22,23]. It has been suggested that the sub-zero temperatures in the Southern Ocean of Antarctica, which allow for high oxygen saturation, may have contributed to the loss of haemoglobin mutation being spread throughout the Channichthyidae family. A period with low levels of iron in the Southern Ocean has also been suggested as an additional selective pressure for this mutation to thrive [24]. Meanwhile, numerous physiological adaptations have been reported in both Channichthyidae and red-blooded members of the notothenioids, including altered mitochondrial structure and function [25,26,27,28,29], and the expression of antifreeze glycopeptides [30,31,32,33,34].

Due to the mitochondrial dependence on oxygen for aerobic respiration, and the oxygen-binding properties of haemoglobin, hypoxia is an important condition to consider when studying mitochondrial haemoglobin. Hypoxia occurs when the demand for oxygen exceeds the supply, resulting in decreased partial pressures of oxygen within tissues that can ultimately lead to impaired physiological function [35]. The lack of oxygen available for complex IV of the mitochondrial electron transport chain impairs the function of the system, leading to a reduced output of ATP [36]. In order to adapt to this low oxygen and low bioenergetic state, cells undergo metabolic reprogramming, including a reduction in the activity of ATP-consuming processes and a switch to anaerobic ATP production [35,37]. The main driver of the eukaryotic response is the well-characterised hypoxia inducible factor (HIF) response, while other pathways, such as the unfolded protein response (UPR) and the mechanistic target of rapamycin (mTOR), are also implicated [38].

Studies of mitochondrial haemoglobin have reported its localisation to the inner mitochondrial membrane [14] and that overexpression of haemoglobin in MN9D cells causes changes to the expression of oxygen homeostasis and mitochondrial oxidative phosphorylation genes [11]. A mechanism by which the ATP synthase regulatory protein inhibitory factor 1 (IF1) controls the synthesis of haemoglobin, due to the pH dependence of the haem synthesis pathway, has also been reported [39]. In this study, we therefore sought to investigate the physiological role of mitochondrial haemoglobin, including in bioenergetics and oxygen homeostasis. Through investigations across multiple species, using both molecular and bioinformatic methods, we examined the conserved role of this novel haemoglobin protein.

## 2. Methods

### 2.1. Rat and Mouse Hypoxia

The samples used to study the impact of hypoxia on mitochondria haemoglobin in rodents were kindly provided by Andrew Murray (University of Cambridge, UK). Rat liver, mouse liver, and mouse quadriceps were collected from animals exposed to hypoxia as described in detail previously by Murray and colleagues [40,41,42]. Briefly, rats and mice were housed in conventional cages in a temperature (23 °C) and humidity-controlled environment with a 12 h/12 h light/dark cycle. They were fed a standard diet and had access to water ad libitum. For the experiment, the rodents either remained under normoxic conditions (21% O_2_) or were housed in hypoxic conditions (10% O_2_) in a flexible-film chamber (PFI Systems Ltd., Milton Keynes, UK). The rats were housed under hypoxic conditions for 14 days, and the mice for 28 days, with 20 air changes/h.

### 2.2. D. melanogaster Hypoxia

Mixed populations of wild-type flies were subjected to the following conditions: 2.5% O_2_ for 30 min at 25 °C followed by normoxia for 30 min at 25 °C, then either stored frozen at −80 °C or subjected to a second cycle of 2.5% O_2_ 30 min at 25 °C, followed by normoxia for 30 min at 25 °C before freezing at −80 °C.

### 2.3. Fish Liver Lysate Preparation

First, 10 mg of liver tissue from each of the fish species (C. rastrospinosus, T. bernacchii, N. rossii and C. gunnari) was added to 100 µL of extraction buffer (1X IP, Dynabeads™ Co-Immunoprecipitation Kit (ThermoFisher, Waltham, MA, USA)) and mechanically homogenised for 1 min with a 1.2–2.0 mL Eppendorf micro-pestle (Sigma-Aldrich, St. Louis, MO, USA). The homogenate was cooled on ice for 15 min before centrifugation at 15,000× *g* for 5 min to pellet the insoluble fraction. The supernatant was collected to be used in the co-immunoprecipitation reaction, and the pellet was discarded.

### 2.4. HEPG2 Treatment with Atractyloside

HEPG2 cells were seeded in T175 flasks (DMEM, 10% FCS, L-glutamine). After 24 h, cells were treated with 8 μM atractyloside for 24 h, following the previous literature [43]. After 24 h of treatment, cells were trypsinised and frozen at −80 °C until use.

### 2.5. HEPG2 Mitochondrial Isolation

Cells were resuspended in mitochondrial extraction buffer and then passed 10 times through a 1 mL syringe with a 26-gauge needle for lysis, as per the published protocols [44]. The homogenate was centrifuged at 2000× *g* for 10 min to clear debris. This step was repeated to clear any remaining nuclear material. The post-nuclear supernatant was then centrifuged at 14,000× *g* for 30 min, and the subsequent supernatant was removed leaving the mitochondrial pellet.

### 2.6. Rat and Mouse Tissue Lysate Preparation

First, 10 mg of either liver or quadriceps muscle tissue was homogenised for 1 min, using a 1.2–2.0 mL Eppendorf micro-pestle (Sigma-Aldrich) in 100 µL of extraction buffer (1X IP, Dynabeads™ Co-Immunoprecipitation Kit (ThermoFisher)). The homogenate was cooled on ice for 15 min before centrifugation at 15,000× *g* for 5 min to pellet the insoluble fraction. The supernatant was collected for the co-immunoprecipitation reaction, and the insoluble fraction was discarded.

### 2.7. D. melanogaster Mitochondrial Isolation

For the isolation, 100 frozen *D. melanogaster* were added to 500 µL of mitochondrial extraction buffer and homogenised for 1 min with a 1.2–2.0 mL Eppendorf micro-pestle (Sigma-Aldrich). The homogenate was spun at 850× *g* for 10 min to clear debris, and the supernatant was then collected and spun at 1000× *g* for a further 10 min to produce a nuclear pellet. The subsequent supernatant was collected, and a final spin was conducted at 12,000× *g* for 30 min to produce a mitochondrial pellet and cytoplasmic supernatant, which were extracted and stored.

### 2.8. Co-Immunoprecipitation (Dynabeads™ Co-Immunoprecipitation Kit (ThermoFisher))

To begin, 10 µg of antibody (anti-HbB, ab227552) was bound to 2 mg of Dynabeads, following the manufacturer’s protocol. The antibody-conjugated Dynabeads were incubated overnight (18 h, 4 °C) with the prepared Notothenioid liver lysate. The Dynabeads were then washed, and the co-immunoprecipitated proteins were eluted and stored at −80 °C.

### 2.9. SDS-PAGE

First, 1 µL of protein sample was added to 5µL of PBS, 3 µL LDS, and 3µL DTT. The solution was boiled at 95 °C for 10 min and loaded onto SDS-polyacrylamide pre-cast gels (NuPAGE™ 4 to 12%, Bis-Tris, 1.0 mm, Mini Protein Gel, 12-well). In preparation for label-free mass spectrometry of co-IP fractions, a fixed voltage (200V) was applied until the dye front had run 2 cm into the gel. The gel was then removed from the casing and stained overnight (18 h, RT) with ProtoBlue Safe Coomassie G-250. In the case of Western blots or MALDI-TOF-MS, the fixed voltage (200V) was applied for 35 min, before continuation of the Western blot protocol or Coomassie staining. Coomassie-stained gels were destained by washing for 10 min with deionised H_2_O three times. The Coomassie-stained sections of the gel lanes were excised, and they were then stored in 1.5 mL Eppendorf tubes (4 °C) until they were ready to be sent for mass spectrometry analysis.

### 2.10. Western Blot

Proteins were transferred from the polyacrylamide gel to a nitrocellulose membrane at 30 V for 60 min. Then, 5 mL 3% (*w*/*v*) milk powder (Marvel) in TBS-T was used to block the membrane for 60 min with gentle agitation (RT). The membrane was then incubated (18 h, 4 °C) with the primary antibody (anti-HbA, ab82871; anti-HbB, ab227552; anti-Beta actin, ab8227; anti-ATP5A, ab245580; anti-ATPIF1, SAB2100188; anti-GAPDH, ab9485) at a 1:5000 dilution in 3% (*w*/*v*) milk powder in TBS-T. The membrane was then washed three times with TBS-T before incubation with the secondary antibody (goat anti-rabbit HRP conjugate, ab6721) at a 1:5000 dilution in 5% (*w*/*v*) milk powder in TBS-T. The membrane was washed another three times with TBS-T before a 5-minute incubation with the ECL substrate and chemiluminescence measurement. Band densities were measured using Image J, and samples were normalised to appropriate loading controls. Statistical analysis was performed in GraphPad Prism.

### 2.11. Label-Free Mass Spectrometry

Samples were analysed by the Centre of Excellence in Mass Spectrometry at the University of York. Protein was in-gel digested post reduction and alkylation. Extracted peptides were analysed over 1 h LC-MS acquisitions with elution from a 50 cm, C18 PepMap column onto a Thermo Orbitrap Fusion Tribrid mass spectrometer using a Waters mClass UPLC. Data analysis was performed using PEAKS StudioX-Pro, employing the Spider search function to include single amino acid point mutations as variable modifications, allowing for better matching to more divergent sequence data [45].

NCBI entries for Trematomus bernacchii (41,453), Notothenia rossii (94), Ommatophoca rossii (65), Erebia rossii (27), Fannyella rossii (17), Anser rossii (12), rossii (50), Chionodraco rastrospinosus (174), and Chaenocephalus aceratus (224) were downloaded into a concatenated database for searching within PEAKS. To qualitatively determine presence of a protein, protein identities were filtered to achieve <1% false discovery rate (FDR) as assessed against a reverse database. Identities were further filtered to require a minimum of two unique peptides per protein group. For quantitative analysis of protein abundance between species, the mapped peptide ion areas were tested using a multi-way ANOVA.

### 2.12. Co-Immunoprecipitation (Invitrogen™ Dynabeads™ Protein G Immunoprecipitation Kit)

First, 10 µg of antibody (anti-HbB, ab227552) was conjugated to 2 mg of Dynabeads (Invitrogen™ Dynabeads™ Protein G Immunoprecipitation Kit), as per the manufacturer’s instructions. The antibody-conjugated Dynabeads were incubated overnight (18 h, 4 °C) with the *D. melanogaster* mitochondrial fraction. The following day, the Dynabeads were washed, and the antibody–antigen complex was eluted and stored at −80 °C.

### 2.13. MALDI-TOF-MS

Samples were analysed by the Centre of Excellence in Mass Spectrometry at the University of York. Following in-gel digestion, peptides were analysed over 20 min acquisitions with elution from a 10 cm Waters T3 nano C18 column onto a Bruker maXis qTOF operated in DDA mode. The resulting peptide spectra were searched against provided sequences and the appropriate species-specific entries in the SwissProt protein database using Mascot.

### 2.14. Molecular Docking

The IF1 3D structure was extracted from its X-ray-resolved complex with the ATP synthase complex from *B. taurus* (cow) (PDB ID:1OHH) and was used for docking simulations (*B. taurus* is the only verified structure from mammals available). The haemoglobin structure was taken from the 3D structure of haemoglobin from *B. taurus* to keep the consistency in the choice of organism and was prepared for docking as tetrameric, dimeric, and monomeric forms (only HbA). All protein structures were minimised before proceeding with docking using Chimera [46].

Patch dock protein–protein docking [47] (bioinfo3d.cs.tau.ac.il/PatchDock/) server was used for different dockings as follows: (A) IF1 to ATPase complex. (B) IF1 monomer to haemoglobin tetramer no ligands. (C) IF1 dimer to haemoglobin tetramer no ligands. (D) IF1 monomer with haemoglobin dimer. (E) IF1 dimer to haemoglobin dimer. (F) IF1 with HbA chains A and C. (G) Voxelotor inhibitor for haemoglobin. The results obtained from this server were further improved with submission to associate refinement server FireDock [48]. The poses were selected based on global energy.

The PDBsum was used to analyse interacting residues in the docked proteins (residue colours based on their properties and the coloured lines joining these residues representing the type of interaction) [49]. The docked proteins were analysed and visualised using ChimeraPyMOL (Version 1.2r3pre.) and UCSF Chimera [46]. The proteins were docked using different webservers for comparison, HDOCK [50] and ClusPro [51] (Appendix A).

### 2.15. Molecular Dynamics of Protein–Protein Complexes

Protein–protein complexes were prepared for MD simulation with GROMACS (‘S2352711015000059’, version2021.5/11/2021) by generating topology files and coordinate files. Simulation boxes of 11.88 × 11.88 × 11.88 nm (haemoglobin tetramer- IF1), 11.89 × 11.89 × 11.89 nm (HbA–IF1) were set with the protein complex centred inside. spc216.gro solvent configuration was used to add water molecules together with the addition of NaCl to neutralize the entire system. The resulting models were parametrised using AMBER99SB-ILDNP Force Field [52,53] implemented in the GROMACS 5.0 software package.

An energy minimisation protocol of 50,000 cycles with the Steepest Descent minimisation algorithm was then applied. Velocity was generated using a V-rescale thermostat, according to a Maxwell distribution at 310 K, with a short 200 ps run in the NVT ensemble. Position restraints were applied to the protein with a force constant of 1000 kJ mol^−1^, for the whole NVT run. The accurate leap-frog algorithm was used as the MD integrator for the whole dynamics using 0.002 fs as the time step for integration. Periodic boundary conditions were applied in all directions (pbc = XYZ). The LINCS algorithm was used to constrain the stretching of all bonds.

Electrostatic interactions were calculated applying the Particle Mesh Ewald (PME) algorithm and a cut-off at 1.2 nm. The same cut-off (1.2 nm) was also chosen for switching off the van der Waals potential. Changing the ensemble to the NPT, we required an 8 ns equilibration run with the Parrinello–Rahman algorithm for pressure coupling under isotropic conditions, and V-rescale was still used for temperature coupling. MD analysis was mostly performed using Pymol [54] and CHIMERA [46]. We used a similar approach to the docking as described in the protein–protein interaction of human glyoxalase II paper [55].

### 2.16. Structure and Sequence Alignment

Sequence alignment was completed with T-Coffee [56]. Structural alignment was completed with UCSF ChimeraX [57], with structures downloaded from PDB for *H. sapiens* haemoglobin b [58] and *D. melanogaster* globin 1 [59].

## 3. Results

### 3.1. Co-Immunoprecipitation of Haemoglobin with ATP Synthase

We first sought to understand the different interaction partners of haemoglobin that are found within the mitochondria through immunoprecipitation studies. A previous immunogold labelling transmission electron microscopy study has shown haemoglobin to be present in proximity to the inner mitochondrial membrane, and its presence was detected by Western blot of the inner mitochondrial membrane subfraction. Others suggest that its presence within certain cell types is physiologically linked to oxidative phosphorylation, and we used anti-ATP5A antibodies to co-immunoprecipitate proteins from homogenised Notothenioid liver tissue lysates. Label-free mass spectrometry analysis of the eluted fractions showed that haemoglobin subunits were precipitated in the red-blooded Notothenioid species *N. rossii* and *T. bernacchii* (Table 1). We then investigated this potential interaction in mammals (rats) and observed that haemoglobin beta co-immunoprecipitated with anti-ATP5A antibodies, which were incubated with mitochondria isolated from rat liver tissue (Table 1, Appendix A).

To confirm whether this interaction was also conserved in non-vertebrates that express haemoglobin, we used an anti-haemoglobin beta antibody in a co-immunoprecipitation reaction with mitochondria isolated from *D. melanogaster.* MALDI-TOF/MS analysis of the co-immunoprecipitation elution revealed that several ATP synthase proteins, as well as the ADP/ATP carrier protein, immunoprecipitated with the anti-HbB antibody (Table 1). Based upon structural and sequence alignment analyses (Appendix A), as well as the highly limited tissue- and sex-specific expression of Glob2 and Glob3 in *D. melanogaster*, we concluded that the anti-HbB antibody was binding to Glob1.

### 3.2. In Silico Interaction between IF1 and Haemoglobin α_2_β_2_

IF1 has previously been reported to regulate haem synthesis [39], and thus haemoglobin levels, so we sought to investigate any potential interaction between the two proteins. As haemoglobin alpha immunoprecipitated with anti-IF1 in red-blooded notothenioid liver tissue (Table 1), we modelled the potential physical interaction between mitochondrial haemoglobin and IF1 through simulating molecular docking interactions (using a combination of Patchdock and then Firedock for refinement).

Table 2 contains the geometric score, interface area size, and normalised desolvation energy from PatchDock, and subsequent FireDock refinement for the docking of IF1 with tetrameric haemoglobin. The tetrameric haemoglobin docking simulation used a minimised structure of haemoglobin with all chains and no ligands, and the docking simulation with haemoglobin alpha used chain C. The docking of the known haemoglobin inhibitor, Voxelotor, was also simulated with PatchDock to compare it to the docking with IF1.

For docking the IF1 monomer to haemoglobin tetramer, we chose 50 docking conformations ranked based on their PatchDock scores and then submitted these to Firedock refinement. The top 10 were selected based on global energy (Appendix A). After assessing the structures using Ramachandran plot to validate the docking pose, we chose FireDock solution number 1 (solution 8 of PatchDock); this conformation had high global energy, attractive van der Waals, and low repulsive van der Waals (Table 2).

The possible interaction of monomeric IF1 with tetrameric haemoglobin according to PatchDock solution 8 was assessed, with all simulations at a constant pH of 7.4 (Figure 1). The energy-stable computational structures produced by PatchDock showed residues 35–60 of IF1 interacting with tetrameric haemoglobin residues.

### 3.3. In Silico Interaction between IF1 and Haemoglobin α

When docking only the alpha globin peptide of haemoglobin (Chain C) with IF1 (Chain A) (Table 3), it showed two polar contacts between the two molecules (5 Å); between residues Glu66 of IF1 protein and Lys99 of haemoglobin α (bond length 2.7 Å), and between His56 of IF1 and Ser138 of haemoglobin α (bond length 3.5 Å) (Figure 2). The isolated haemoglobin alpha chain interacted with residues 52–74 (C-terminal end) of IF1. The interacting residues common to both haemoglobin α_2_β_2_ and haemoglobin α were Val1, Leu2, Asn131, Thr134, Thr137, and Ser 138 (Appendix A). This interaction was confirmed experimentally by co-immunoprecipitation of haemoglobin α with IF1 in *N. rossii* (Table 1).

### 3.4. Molecular Docking Simulation of IF1 and Haemoglobin α_2_β_2_

The best docked conformation for haemoglobin α_2_β_2_ was chosen according to the global energy of the docked compound (Table 2), and a molecular docking interaction was simulated (Figure 3). The docked protein was prepared creating a topology file that contains all the information of the structure, and AMBER99SB-ILDN (Lindorff-Larsen) forcefield was applied on the complex [60].

The system was equilibrated at 305K constant temperature, density, and pressure; the structure was relaxed using energy minimisation (Appendix A). The quality of simulated structure was checked using root-mean-square deviation over the simulation time (8 ns), which is a common technique to verify the stability of MD simulation. The system was stable between 4 and 7 ns (Figure 4).

Most of the interactions that were present at the start of the MD simulation were observed at the end of the MD stabilisation (Appendix A). An additional hydrogen bond was seen between Asn131 and Lys46, with a measured distance of 2.1 Å. Ser81 of haemoglobin was seen to form a hydrogen bond with Lys39 both before and after MD stabilisation. Another polar contact was observed between Leu2 of haemoglobin with Glu50 of IF1, with a measured distance of 2.1 Å.

### 3.5. Mitochondrial Haemoglobin, ATP5A, and IF1 Expression in Rodents Exposed to Chronic Hypoxia

As a previous study of non-erythroid haemoglobin linked its expression with mitochondrial function, via modulation of oxidative phosphorylation and oxygen homeostasis genes [11], we characterised the expression of mitochondrial haemoglobin subunits alpha and beta, ATP5A of the ATP synthase enzyme, and IF1 in response to hypoxia. The expression varied according to species, tissue type, and hypoxia conditions (Figure 5, Appendix A). No significant changes in expression were observed in rat liver in response to hypoxia, while in mouse liver, only IF1 showed a significant increase in expression in response to hypoxia. However, in mouse quadriceps muscle, haemoglobin alpha expression was only detected after exposure to hypoxia, and haemoglobin beta expression was significantly increased above normoxic levels in response to hypoxia.

### 3.6. Mitochondrial Haemoglobin, ATP5A, and IF1 Expression in D. melanogaster Exposed to Acute Hypoxia Cycles

To then investigate the impact of hypoxia on mitochondrial haemoglobin and the ATP synthase machinery in an invertebrate species, *D. melanogaster* were exposed to acute cycles of hypoxia, and mitochondria were then isolated from whole *D. melanogaster*. The expression of their mitochondrial haemoglobin, ATP5A, and IF1 was measured (Figure 6, Appendix A). After two cycles of hypoxia and recovery, mitochondrial haemoglobin, IF1, and ATP5A expression were found to be expressed at significantly higher levels.

### 3.7. HEPG2 Cells Treated with Atractyloside Show an Increase in Haemoglobin α and β Expression

To investigate whether the dependence of mitochondrial haemoglobin expression was dependent on oxygen availability or ATP levels, HEPG2 cells were treated for 24 h with atractyloside, an inhibitor of the ADP/ATP translocase that causes a reduction in mitochondrial ATP synthesis. We observed a trend toward increased in the expression of both haemoglobin α and β subunits (Figure 7, Appendix A).

## 4. Discussion

### 4.1. Haemoglobin Binds to ATP Synthase and Associated Proteins

We found evidence of interactions between mitochondrial haemoglobin and ATP synthase, ADP/ATP translocase, and IF1 that are conserved across different species. Using in silico methods, we described a binding interaction between haemoglobin and IF1. Our findings concur with previous observations of haemoglobin being localised to the inner mitochondrial membrane by Shephard et al. [14]. Additionally, a co-immunoprecipitation reaction using an anti-haemoglobin beta antibody with total cell extracts from MS patient motor cortices pulled down ATP synthase subunits alpha and beta, ADP/ATP translocase 4, and the mitochondrial phosphate carrier [61]. Beyond binding to the ATP-associated proteins of the mitochondria, the binding interaction of haemoglobin and organic phosphates (such as ATP and ADP) is a well-characterised phenomenon [2,3,4,62].

### 4.2. ATP Synthase Activity and Haemoglobin

A study of barley aleurone tissue investigated the effects of different respiratory inhibitors on whole-cell haemoglobin expression, and found that, in response to inhibitors that reduced oxygen consumption, as well as to uncouplers that increased oxygen consumption, haemoglobin expression was upregulated [63]. The authors suggested that haemoglobin expression was not directly responsive to oxygen usage but rather to ATP availability in the tissue. To understand whether mitochondrial haemoglobin was subject to the same trend, we treated HEPG2 cells with atractyloside, an inhibitor of the ADP/ATP translocase that leads to reduced mitochondrial ATP synthesis. We observed a trend toward increased in the expression of mitochondrial haemoglobin α and β. Like the work of Nie et al., this suggests that the expression of mitochondrial haemoglobin could be responsive to ATP levels, not oxygen availability. As HEPG2 is a cell line derived from liver tissue, future studies should investigate cell lines derived from skeletal muscle to see how they might reflect our studies of hypoxic rodent tissue.

Previously, it has been reported that IF1 deficient zebrafish exhibit profound anaemia [39]. The absence of the IF1 allows for the reverse hydrolytic function of ATP synthase to occur, increasing the pH within the mitochondrial matrix, which in turn causes the inhibition of the haem synthetic pathway enzyme ferrochelatase. The absence of haem leads to a reduced transcription of the globin genes, and thus anaemia. This suggests there is a potential mechanism for the modulation of haemoglobin synthesis in response to the bioenergetic state of the mitochondria.

Hypoxia is characterised by low ATP and acidosis, which alters mitochondrial structure and function [64,65,66]. Our observed localisation of mitochondrial haemoglobin to the ATP synthase machinery and associated proteins suggests that the link between mitochondrial haemoglobin and ATP synthase may be involved in the hypoxia response.

### 4.3. Mitochondrial Haemoglobin Abundance Is Responsive to Hypoxia

We observed that mitochondrial haemoglobin was upregulated in response to hypoxia in the skeletal muscle of mice and in *D. melanogaster*. Exercise leads to a drop in oxygen concentration in skeletal muscle [67,68], and thus hypoxia, and so we see a different specific response in skeletal muscle compared with the liver with hypoxia. The consistent expression levels of mitochondrial haemoglobin in the liver tissue in response to hypoxia might suggest that the hepatic mitochondrial hypoxia response does not require upregulated haemoglobin, perhaps due to other tissue-specific mechanisms [40]. *D. melanogaster* have a well-described tolerance to hypoxia [69,70], and the modulation of mitochondrial haemoglobin content appears to be a part of this response. Future studies should consider the tissue-specific haemoglobin expression in response to hypoxia in *D. melanogaster*, while noting the tissue-specific expression of *D. melanogaster* haemoglobin.

Functional studies of mitochondrial haemoglobin include the NO-consuming yeast flavohaemoglobin (YHb), where it has been reported that YHb is distributed between the cytosol and the mitochondria, whereas all of the YHb was located in the mitochondrial fraction of JM43 cells grown in anaerobic conditions [71]. A study of U937 cells treated with differing concentrations of haemoglobin found that intracellular haemoglobin co-localises with mitochondrial complex I, stimulates mitochondrial respiration, and leads to increased expression of HIF1a and Nrf2 mRNA [72]. Treatment of nigral dopaminergic neurons with the complex I inhibitor rotenone led to a significant decrease in haemoglobin a and b mRNA levels [12]. When viewed in the context of the data we report here, this would indicate that the presence of mitochondrial haemoglobin is not dependent upon electron transport chain activity.

## 5. Conclusions

The role of mitochondrial haemoglobin is not yet known, though there is a suggestion that it may provide a protective response against oxidative stress. We present evidence of the interaction between mitochondrial haemoglobin and ATP synthase proteins, as well as showing that it has a modulated expression in response to hypoxia. Moreover, our observations fit in with previously described mechanisms that have shown that hypoxia alters mitochondrial function via acidosis; further, we provide evidence to support the link between haemoglobin expression and ATP availability. We have modelled a mechanism by which the IF1-regulated pH of the mitochondrial matrix regulates haemoglobin synthesis. The studies we present here directly link the activity of mitochondrial ATP synthase with the key protein for oxygen delivery in metazoans.

## Figures and Tables

**Figure 1 cells-12-00912-f001:**
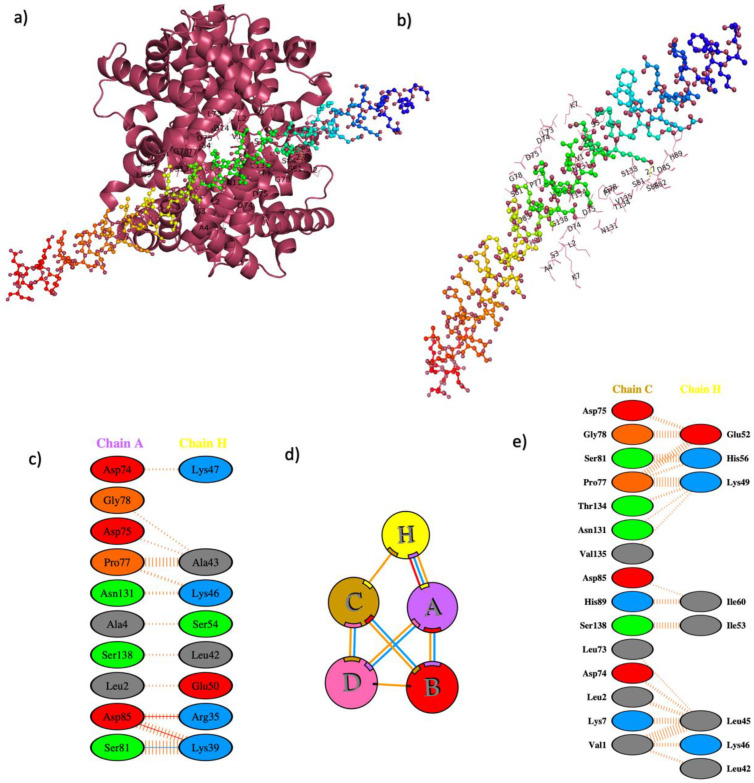
**Molecular docking of IF1 and tetrameric haemoglobin, and a visualisation of interacting residues.** (**a**) Docking of tetrameric haemoglobin (red) with a single chain of IF1; (**b**) the interacting residues of IF1 in haemoglobin docking simulation; (**c**) schematic representation of interacting residues between haemoglobin chain A and IF1 chain; (**d**) positioning of IF1 in proximity to interacting haemoglobin chains A and C, where different coloured lines represent different interactions between amino acids, specifically: red—salt bridges, yellow—disulphide bonds, blue—hydrogen bonds, and orange—non-bonded contacts; (**e**) schematic representation of the interacting residues between haemoglobin chain C and IF1 chain.

**Figure 2 cells-12-00912-f002:**
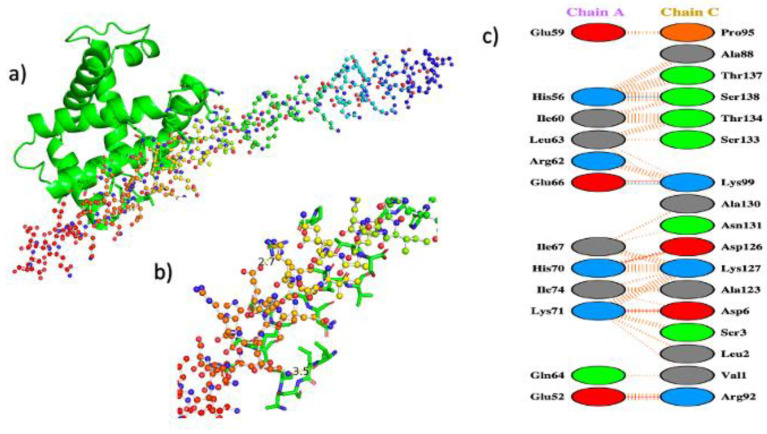
**Molecular docking of IF1 and monomeric haemoglobin alpha, and a visualisation of interacting residues.** (**a**) Docking of monomeric haemoglobin alpha (green) with a single chain of IF1; (**b**) the monomeric haemoglobin alpha interaction site of IF1 chain; (**c**) schematic representation of the interacting residues between monomeric haemoglobin alpha and IF1 chain.

**Figure 3 cells-12-00912-f003:**
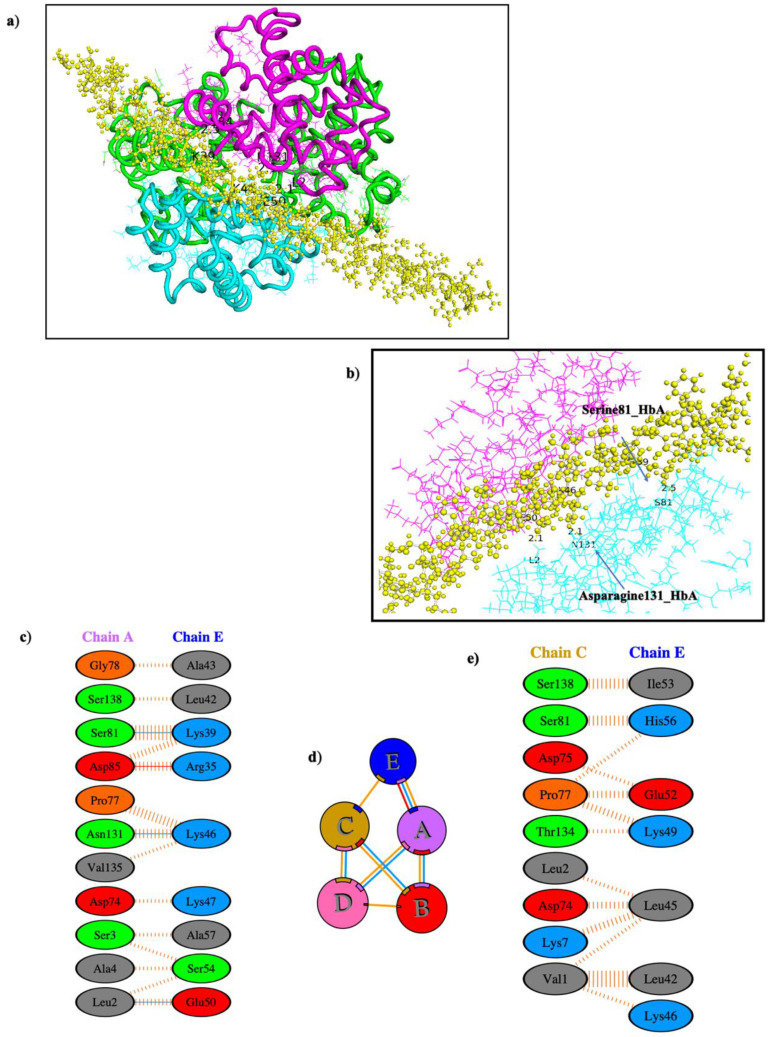
**Haemoglobin–IF1 association after 8 ns MD simulation**. (**a**) IF1 docked in the cleft of two alpha chains of haemoglobin (pink—Chain A, light blue—Chain C, yellow—IF1). (**b**) Shows interaction of haemoglobin residues with IF1 residues—Serine 84 and Asparagine 131 of HbA. Interacting residues. (**c**) Schematic representation of interacting residues between tetrameric haemoglobin chain A and IF1 after 8 ns MD simulation. (**d**) Positioning of IF1 in proximity to interacting haemoglobin chains A and C after 8 ns MD simulation, where different coloured lines represent different interactions between amino acids, specifically: red—salt bridges, yellow—disulphide bonds, blue—hydrogen bonds, and orange—non-bonded contacts. (**e**) Schematic representation of interacting residues between tetrameric haemoglobin chain C and IF1 after 8 ns MD simulation.

**Figure 4 cells-12-00912-f004:**
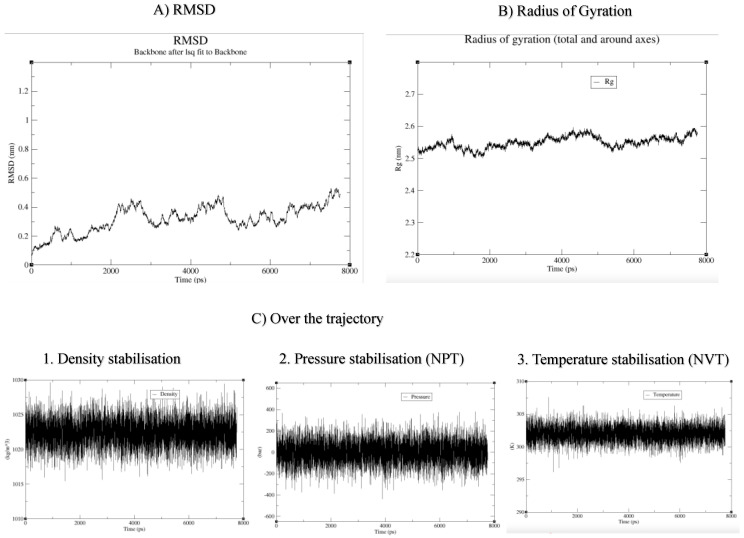
**Quantitative quality check for the MD run.** (**A**) Root-mean-square deviation (RMSD); (**B**) radius of gyration shows a reasonable invariant Rg values across the MD run of 8 ns, indicating the protein remains very stable; (**C**) the thermodynamic factors of density, pressure, and temperature are stable across the MD run.

**Figure 5 cells-12-00912-f005:**
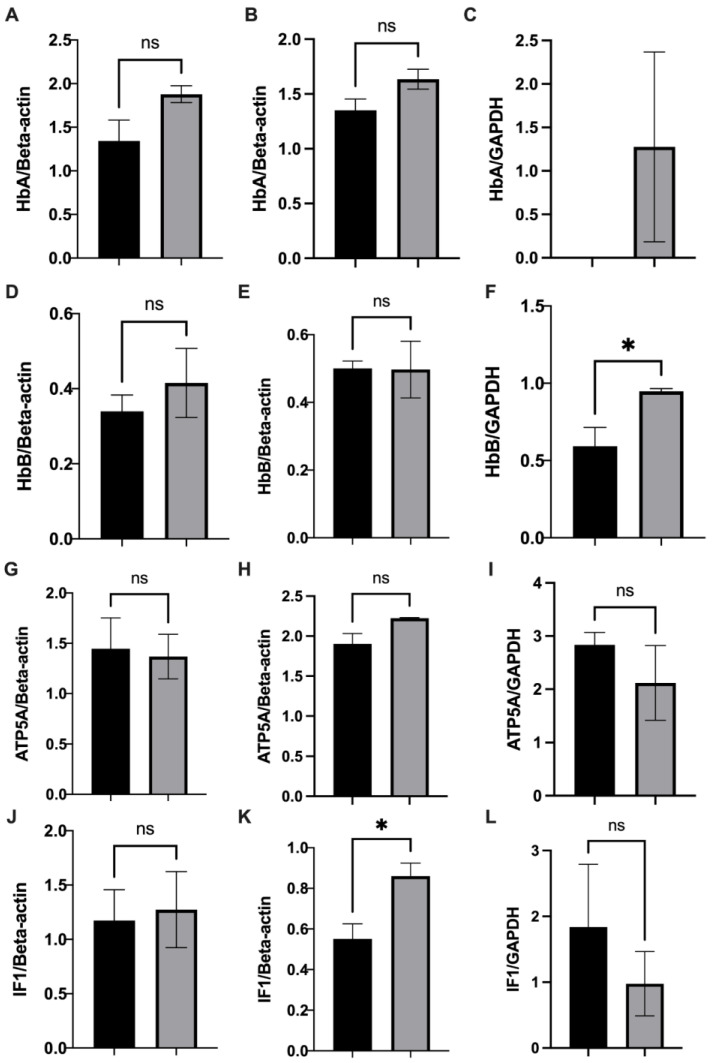
Relative mitochondrial protein expression in normoxic (black) and hypoxic (grey) (**A**,**D**,**G**,**J**) rat liver,(**B**,**E**,**H**,**K**) mouse liver, and (**C**,**F**,**I**,**L**) mouse quadriceps muscle. (**A**–**C**) haemoglobin α; (**D**–**F**) haemoglobin β; (**G**–**I**) ATP5A; (**J**–**L**) IF1. Liver mitochondria protein expression normalised to β-actin and quadriceps muscle mitochondria protein expression normalised to GAPDH (see Appendix A). Data presented as mean with SEM, * *p* < 0.05, N = 3, Student’s unpaired *t*-test.

**Figure 6 cells-12-00912-f006:**
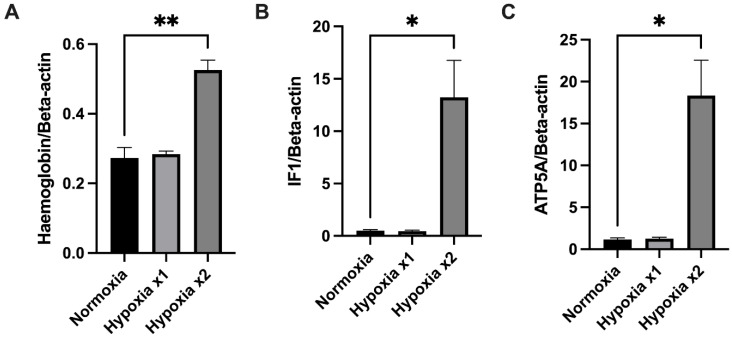
**Relative expression of mitochondrial proteins in normoxic and hypoxic D. melanogaster.** (**A**) haemoglobin; (**B**) IF1; (**C**) ATP5A. Protein expression normalised to β-actin (see Appendix A). Data are presented as mean with SEM, * *p* < 0.05, ** *p* < 0.005 N = 3, Student’s unpaired *t*-test.

**Figure 7 cells-12-00912-f007:**
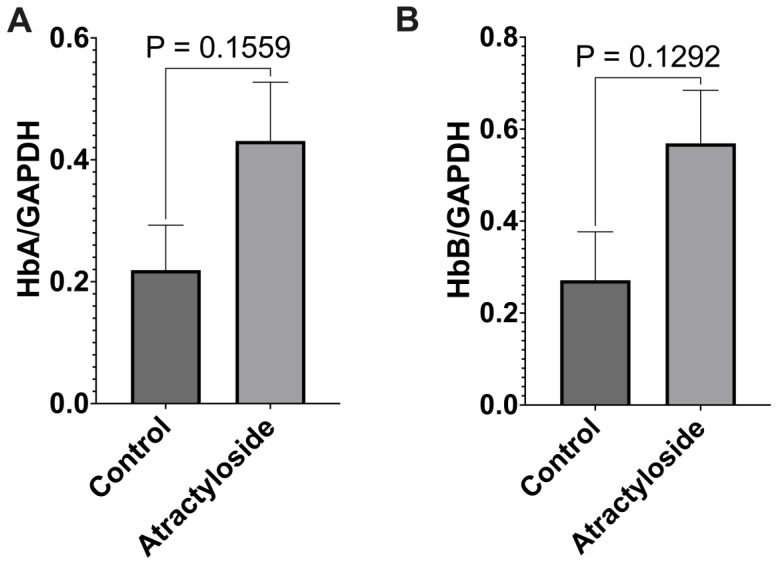
**Atractyloside-treated HEPG2 cells show a trend toward increased expression of mitochondrial haemoglobin.** (**A**) Relative expression of haemoglobin α in mitochondria isolated from atractyloside-treated HEPG2 cells. (**B**) Relative expression of haemoglobin β in mitochondria isolated from atractyloside-treated HEPG2 cells. Data are presented as mean with SEM, N = 3, student’s unpaired *t*-test.

**Table 1 cells-12-00912-t001:** Co-immunoprecipitation of haemoglobin and ATP synthase proteins. Haemoglobin subunits co-immunoprecipitate with ATP synthase subunit a in rat liver lysates and notothenioid liver lysates. Haemoglobin co-immunoprecipitates with IF1 in notothenioid liver lysates. Multiple ATP subunits immunoprecipitate with haemoglobin in D. melanogaster.

Purification Method	Antibody	Sample	Detection Method	Identified Proteins
Dynabeads™ Co-Immunoprecipitation Kit	Anti-ATP5A	Rat liver lysate	Western Blot	Haemoglobin beta
		Notothenioid liver lysate	Label-free mass spectrometry	Haemoglobin alpha, haemoglobin beta
	Anti-IF1	Notothenioid liver lysate	Western Blot	Haemoglobin alpha
Invitrogen™ Dynabeads™ Protein G Immunoprecipitation Kit	Anti-HbB	Drosophila mitochondria	MALDI-TOF/MS	ATP synthase subunit alpha, ATP synthase subunit beta, ADP/ATP carrier protein, ATP synthase subunit d, ATP synthase subunit gamma, ATP synthase subunit b, putative ATP synthase subunit f

**Table 2 cells-12-00912-t002:** Docking scores from PatchDock and FireDock for tetrameric haemoglobin and IF1. Binding affinity (score) and binding energy (global energy) of the solution in kcal/mol; attractive and repulsive Van der Waals forces (kJ/mol) and atomic contact energy (kcal/mol).

SolutionNumber (FireDock)	Solution Number (PatchDock)	Global Energy (kcal/mol)	Attractive VdW (kcal/mol)	Repulsive VdW (kcal/mol)	ACE (kcal/mol)	Score	Area
1	8	−37.84	−27.85	13.77	3.13	11,588	1356.50
2	3	−20.75	−18.22	6.18	2.50	11,988	1706.30
3	34	−19.64	−33.34	24.33	10.24	10,584	1428.20
4	25	−10.15	−28.12	15.34	6.04	10,824	1323.70
5	43	−5.36	−37.07	30.53	18.52	10,296	1449.80
6	31	−5.14	−27.18	7.12	12.45	10,608	1849.70
7	2	−3.89	−28.98	7.95	5.50	12,304	1659.30
8	5	−2.64	−18.48	2.52	6.48	11,966	1606.50

**Table 3 cells-12-00912-t003:** Docking scores from PatchDock and FireDock for monomeric haemoglobin alpha and IF1- binding affinity (score) and -binding energy (global energy) of the solution in kcal/mol; attractive and repulsive Van der Waals forces (kJ/mol) and atomic contact energy (kcal/mol).

Solution Number (FireDock)	Solution Number (PatchDock)	Global Energy (kcal/mol)	Attractive VdW (kcal/mol)	Repulsive VdW (kcal/mol)	ACE (kcal/mol)	Score	Area
1	30	−27.01	29.93	11.49	7.74	8696	1125.90
2	19	−16.10	−25.88	13.35	11.48	8852	1081.70
3	44	−15.84	−28.74	26.23	5.54	8340	1267.10
4	15	−12.18	−29.70	13.98	8.57	9022	1087.50
5	38	−8.08	−18.35	14.55	5.09	8464	1444.70
6	7	−6.67	−33.25	18.06	15.48	9368	1154.40
7	11	−4.14	−6.10	2.72	4.76	9132	1123.70
8	25	−0.54	−26.40	13.84	10.61	8814	1202.00
9	36	−0.33	−19.90	7.34	4.43	8502	1368.90

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
