# Peer review of "Mitochondrial Haemoglobin Is Upregulated with Hypoxia in Skeletal Muscle and Has a Conserved Interaction with ATP Synthase and Inhibitory Factor 1"

_cells, 2023, doi:10.3390/cells12060912_

Round 1

Reviewer 1 Report (Previous Reviewer 3)

The manuscript has been improved from the previous version. In the current form, in my opinion, is suitable for publication.

Reviewer 2 Report (Previous Reviewer 2)

The authors did not address the major concerns that their analysis of protein levels do not prove their claims. All they have done is clarify particular confusions with their manuscript without addressing this issue.

Reviewer 3 Report (Previous Reviewer 1)

I find the manuscript of great interest, understand the changes made that improved the manuscript and consider it adequate. I have no remarks.

This manuscript is a resubmission of an earlier submission. The following is a list of the peer review reports and author responses from that submission.

Round 1

Reviewer 1 Report

The authors are presenting a very interesting translational (interspecies) analysis of mitochondrial haemoglobin in relation of hypoxia.

The manuscript has the merit to draw attention to a not so much know particular action of Haemoglobin.

The methodology Is sound, and the manuscript clear.

The methods described are sufficiently detailed to reperform the experiment.

I appreciated the discussion since it is balanced, sufficiently prudent since more is needed 

I have however some suggestions:

Minor:

Line 243 -> A large blank space is present in the manuscript with no real utility. Simple correction needed.

The graphs are useful but need a bit of coherence, the templates are different, the p values are sometimes in plain text (figure, numbers) and sometimes asterisks (*), just choose which you consider the best, I may suggest numbers.

It may be useful, not only to discuss hypoxia but also “relative hypoxia” as it has already been suggested. I would be very interested in the results in the future. (see refs.)

Salvagno M, Coppalini G, Taccone FS, Strapazzon G, Mrakic-Sposta S, Rocco M, Khalife M & Balestra C. (2022). The Normobaric Oxygen Paradox-Hyperoxic Hypoxic Paradox: A Novel Expedient Strategy in Hematopoiesis Clinical Issues. Int J Mol Sci 24.

Leveque C, Mrakic-Sposta S, Lafere P, Vezzoli A, Germonpre P, Beer A, Mievis S, Virgili F, Lambrechts K, Theunissen S, Guerrero F & Balestra C. (2022). Oxidative Stress Response's Kinetics after 60 Minutes at Different (30% or 100%) Normobaric Hyperoxia Exposures. Int J Mol Sci 24.

Thank you for giving me the opportunity of reviewing such an interesting manuscript.

Author Response

Reviewer 1

We thank the reviewer for taking the time to consider our manuscript in such depth and we have made amendments accordingly. We have also endeavoured to explain our rationale in making decisions or presenting results where the reviewer has raised points of interest. The manuscript is undoubtedly improved thanks to their contribution.

Major comments

  1. For co-IP, does this interaction survive treatment with a membrane disruptor? Often proteins will co-IP if they share a membrane. This is true for DNA binding proteins as well. Using a DNase treatment helps discriminate real interactions from those that are indirect. This may be important as multiple proteins were identified in the IP that are membrane associated.

In this instance we did not use membrane disruptors or DNase treatment, although we are aware of this useful practise in some protocols used for co-immunoprecipitation reactions.

Haemoglobin is a soluble, non-membrane protein, and that haemoglobin was found to co-immunoprecipitate with ATP synthase and associated proteins, across a variety of different species, also with soluble, non-membrane protein ATPIF1. We think it is highly unlikely that this interaction is an artefact that arises due to DNA binding protein interactions or as a membrane protein-membrane protein interaction.

  1. For the predicted interaction figures, the definition of the proposed interactions between IF1 and haemoglobin need to be stated in the legends. Specifically, what do the different connecting lines indicate?

We apologise for the omission of this information from the manuscript. The different coloured lines represent different interactions between amino acids, specifically: red- salt bridges, yellow- disulphide bonds, blue-hydrogen bonds, orange-non bonded contacts. This information has now been added to appropriate figure legends.

  1. One main point of the study (as indicated by your title and abstract) is that mitochondrial haemoglobin is upregulated by hypoxia. However, the data are less than convincing. The muscle normoxia samples do indeed look very low although the third repeat appears to be significantly underloaded. However, only one of the hypoxic samples exhibits elevated haemoglobin levels, the other two do not. This needs to be repeated or some explanation provided on why there are such variations. In addition, it should be made clear in the title and abstract that this potential effect was specific for muscle. Finally, were the authors surprised, give the occurrence of hypoxia in muscle cells, that HbA levels were below the limits of detection in these studies?

When asking about the expression of mitochondrial haemoglobin, we assume the reviewer is referring to the haemoglobin a blots, where there is some variation between the amount of sample loaded into each lane.

There is clearly a smaller quantity of sample loaded into lane 3 of the normoxic samples. However, the loading control was well within the limits of detection, and haemoglobin a was also not detected in lanes 1 and 2, so the absence of a band in lane 3 must be considered a biological observation rather than an experimental artefact.

With respect to the hypoxic samples from the mouse quadriceps tissue, the darker haemoglobin a band is due to some additional sample being loaded in comparison with lanes 5 and 6, and though much lighter in contrast, there are bands present for haemoglobin a in lanes 5 and 6.

Following the reviewer’s suggestion, we have amended both the title and abstract to reflect the fact that this change was found in skeletal muscle samples.

In response to the last part of this comment, the surprises are what make the research interesting!  One possibility is that the expression of haemoglobin a increases within skeletal muscle only as a response to hypoxia (e.g. experienced during exercise), and that basal levels of expression are relatively low. The limits of detection with this technique doesn’t necessarily coincide with physiologically important levels of proteins which could be functionally important even at extremely low levels.

  1. The Drosophila studies needs more explanation. Do the authors think that all tissue types exhibit elevated haemoglobin? This would be different than the mammalian studies.

We are not able to define which tissues exhibit the elevated mitochondrial haemoglobin expression in response to hypoxia. However the data would be limited by which cells within the given tissue type contain mitochondria, and whether that tissue type is known to express haemoglobin. In this respect there are studies published by Thomas Hankeln and Thorsten Burmester which have reported the tissue-specific nature of globin expression in Drosophila.

We have added a comment in the results section to make it explicit that this was measured in whole Drosophila, and another comment in the relevant section of the discussion to reflect this (“Future studies should consider the tissue-specific haemoglobin expression in response to hypoxia in D. melanogaster, while noting the tissue-specific expression of D. melanogaster haemoglobin.”).

  1. The authors conclude that increased HbA and HbB expression is observed in HepG2 cells. However, the P values indicate that this is still a question. Also, HepG2 are liver cells. Do the authors conclude that their animal studies are different than cell culture experiments with respect to liver response to hypoxia/ATP depletion? This figure should be omitted unless more information is provided.

To reflect the reviewer’s observation that, while still low, the p-values are above the typical 0.05 significance threshold, we have amended the text within the manuscript to note that there was only a trend toward increased expression. Given that the haemoglobin expression did not reach significance in the Atractyloside treated HEPG2 liver cells (which is concurrent with the hypoxia liver tissue samples), we have added an additional comment in the discussion that suggests further investigation in a skeletal muscle cell line might also reflect the trend we report in the hypoxic tissue samples.

  1. I am unclear about what Figure 8 is testing. The connection between HbB pulling down unknown ATPases and haem hypoxia control needs to be clarified.

The intention behind the experiments represented by figure 8 was to further reinforce the data presented in the earlier co-IP figures, and the in silico investigations. However, we appreciate the reviewer’s comment that this may pull down other unknown ATPase enzymes (not just F-type mitochondrial ATPase). We have therefore made the decision to remove this figure and any reference to it from the manuscript.

Reviewer 2 Report

Ebanks et al.

Summary:

This study investigates a potentially interesting connection between mitochondrial haemoglobin and regulation of ATP production. A role for haemoglobin has been previously suggested in the literature for several mitochondrial maintenance functions although molecular details have been sketchy. The modeling studies provide a starting point to start testing the importance of HbA and IF1. However, additional experiments are required to convincingly prove the connection between these two processes.

Major concerns:

1) For co-IP, does this interaction survive treatment with a membrane disruptor? Often proteins will co-IP if they share a membrane. This is true for DNA binding proteins as well. Using a DNase treatment helps discriminate real interactions from those that are indirect. This may be important as multiple proteins were identified in the IP that are membrane associated.

2) For the predicted interaction figures, the definition of the proposed interactions between IF1 and haemoglobin need to be stated in the legends. Specifically, what do the different connecting lines indicate?

3) One main point of the study (as indicated by your title and abstract) is that mitochondrial haemoglobin is upregulated by hypoxia. However, the data are less than convincing. The muscle normoxia samples do indeed look very low although the third repeat appears to be significantly underloaded. However, only one of the hypoxic samples exhibits elevated haemoglobin levels, the other two do not. This needs to be repeated or some explanation provided on why there are such variations. In addition, it should be made clear in the title and abstract that this potential effect was specific for muscle. Finally, were the authors surprised, give the occurrence of hypoxia in muscle cells, that HbA levels were below the limits of detection in these studies?

4) The Drosophila studies needs more explanation. Do the authors think that all tissue types exhibit elevated haemoglobin? This would be different than the mammalian studies.

5) The authors conclude that increased HbA and HbB expression is observed in HepG2 cells. However, the P values indicate that this is still a question. Also, HepG2 are liver cells. Do the authors conclude that their animal studies are different than cell culture experiments with respect to liver response to hypoxia/ATP depletion? This figure should be omitted unless more information is provided.

6) I am unclear about what Figure 8 is testing. The connection between HbB pulling down unknown ATPases and haem hypoxia control needs to be clarified.

Minor points.

It may be helpful to utilize a different mitochondrial loading control (e.g., Tom20) as although ß-actin and GAPDH can be found in the mitochondria, they are also found in the cytoplasm and may piggyback into the crude mitochondrial preps. This may help reduce the variability seen in some of the experiments.

Although “previously described” and referenced, it would be helpful if a brief description of the hypoxic protocols for the animal studies is presented.

Author Response

  1. It may be helpful to utilize a different mitochondrial loading control (e.g., Tom20) as although ß-actin and GAPDH can be found in the mitochondria, they are also found in the cytoplasm and may piggyback into the crude mitochondrial preps. This may help reduce the variability seen in some of the experiments.

We thank the reviewer for this suggestion as a means of reducing variability seen across samples within blots. Choice of control is always rather complicated, there is something to be said for a control that is informative for whole homogenates and mitochondrial preparations. But justifications can definitely be made both ways.

  1. Although “previously described” and referenced, it would be helpful if a brief description of the hypoxic protocols for the animal studies is presented.

A description of the hypoxic protocols for the animal studies has now been added to the manuscript.

Reviewer 3 Report

The paper of Ebanks and collaborators aims at investigating the interaction between haemoglobin and mitochondrial proteins by using co-immunoprecipitation, in silico and in vitro experiments across multiple species. The paper is interesting and well written especially regarding the in silico analysis. In my opinion representative western blotting images of the graphs of figures 5, 6 and 7 should be shown in the figures. Moreover, according to the statistics, the expression of haemoglobin alpha and beta subunits in atractyloside treated HEPG2 cells does not seem to be increased, likely because of the small number of samples. I suggest that the authors rephrase the sentence or increase the number of the samples to confirm whether the statistically significant difference exists.

Author Response

Thank-you for your review and positive comments.

Minor comments

  1. Line 243 -> A large blank space is present in the manuscript with no
    real utility. Simple correction needed.

This will be removed in the editorial stage by the journal.

  1. The graphs are useful but need a bit of coherence, the templates are
    different, the p values are sometimes in plain text (figure, numbers) 
    and sometimes asterisks (*), just choose which you consider the best, I 
    may suggest numbers.

We have chosen to use asterisks where the p value has reached significance, and the decision to use numbers in figure 7 is to demonstrate the trend toward significance in the results of that experiment, but that the 0.05 threshold had not been crossed. We then chose to leave other, non-significant results without a number, to leave the graphs uncluttered.

  1. It may be useful, not only to discuss hypoxia but also “relative
    hypoxia” as it has already been suggested. I would be very interested in 
    the results in the future. (see refs.)

The concept of ‘relative hypoxia’ and the physiological response to this is an exciting area of research, and we too would be interested to see if the mitochondrial haemoglobin expression response is repeated in response to relative hypoxia of skeletal muscle.

Round 2

Reviewer 2 Report

The authors only responded to a subset of my concerns. In this, they did not address the concerns that the major point of their paper was not convincing due to problems with protein loading. The state "justifications can be made both ways" but they did not provide their justification why there blots were unevenly loaded and why actin represented a good mitochondrial marker. The major concerns in my critique were not mentioned. Was there a problem in the upload of the response?

Author Response

Dear Reviewer.

I must apologise. I think the reviewers comments were labelled differently in the version I received compared with the numbering in the response portal. In order to try and ensure you receive the fullest response I have included the responses to all reviewers here, hopefully you will find your questions answers adequately. With many thanks for your time and attention.

Responses to reviewer comments for Cells manuscript

Reviewer 1

We thank the reviewer for taking the time to consider our manuscript in such depth and we have made amendments accordingly. We have also endeavoured to explain our rationale in making decisions or presenting results where the reviewer has raised points of interest. The manuscript is undoubtedly improved thanks to their contribution.

Major comments

  1. For co-IP, does this interaction survive treatment with a membrane disruptor? Often proteins will co-IP if they share a membrane. This is true for DNA binding proteins as well. Using a DNase treatment helps discriminate real interactions from those that are indirect. This may be important as multiple proteins were identified in the IP that are membrane associated.

In this instance we did not use membrane disruptors or DNase treatment, although we are aware of this useful practise in some protocols used for co-immunoprecipitation reactions.

Haemoglobin is a soluble, non-membrane protein, and that haemoglobin was found to co-immunoprecipitate with ATP synthase and associated proteins, across a variety of different species, also with soluble, non-membrane protein ATPIF1. We think it is highly unlikely that this interaction is an artefact that arises due to DNA binding protein interactions or as a membrane protein-membrane protein interaction.

  1. For the predicted interaction figures, the definition of the proposed interactions between IF1 and haemoglobin need to be stated in the legends. Specifically, what do the different connecting lines indicate?

We apologise for the omission of this information from the manuscript. The different coloured lines represent different interactions between amino acids, specifically: red- salt bridges, yellow- disulphide bonds, blue-hydrogen bonds, orange-non bonded contacts. This information has now been added to appropriate figure legends.

  1. One main point of the study (as indicated by your title and abstract) is that mitochondrial haemoglobin is upregulated by hypoxia. However, the data are less than convincing. The muscle normoxia samples do indeed look very low although the third repeat appears to be significantly underloaded. However, only one of the hypoxic samples exhibits elevated haemoglobin levels, the other two do not. This needs to be repeated or some explanation provided on why there are such variations. In addition, it should be made clear in the title and abstract that this potential effect was specific for muscle. Finally, were the authors surprised, give the occurrence of hypoxia in muscle cells, that HbA levels were below the limits of detection in these studies?

When asking about the expression of mitochondrial haemoglobin, we assume the reviewer is referring to the haemoglobin a blots, where there is some variation between the amount of sample loaded into each lane.

There is clearly a smaller quantity of sample loaded into lane 3 of the normoxic samples. However, the loading control was well within the limits of detection, and haemoglobin a was also not detected in lanes 1 and 2, so the absence of a band in lane 3 must be considered a biological observation rather than an experimental artefact.

With respect to the hypoxic samples from the mouse quadriceps tissue, the darker haemoglobin a band is due to some additional sample being loaded in comparison with lanes 5 and 6, and though much lighter in contrast, there are bands present for haemoglobin a in lanes 5 and 6.

Following the reviewer’s suggestion, we have amended both the title and abstract to reflect the fact that this change was found in skeletal muscle samples.

In response to the last part of this comment, the surprises are what make the research interesting!  One possibility is that the expression of haemoglobin a increases within skeletal muscle only as a response to hypoxia (e.g. experienced during exercise), and that basal levels of expression are relatively low. The limits of detection with this technique doesn’t necessarily coincide with physiologically important levels of proteins which could be functionally important even at extremely low levels.

  1. The Drosophila studies needs more explanation. Do the authors think that all tissue types exhibit elevated haemoglobin? This would be different than the mammalian studies.

We are not able to define which tissues exhibit the elevated mitochondrial haemoglobin expression in response to hypoxia. However the data would be limited by which cells within the given tissue type contain mitochondria, and whether that tissue type is known to express haemoglobin. In this respect there are studies published by Thomas Hankeln and Thorsten Burmester which have reported the tissue-specific nature of globin expression in Drosophila.

We have added a comment in the results section to make it explicit that this was measured in whole Drosophila, and another comment in the relevant section of the discussion to reflect this (“Future studies should consider the tissue-specific haemoglobin expression in response to hypoxia in D. melanogaster, while noting the tissue-specific expression of D. melanogaster haemoglobin.”).

  1. The authors conclude that increased HbA and HbB expression is observed in HepG2 cells. However, the P values indicate that this is still a question. Also, HepG2 are liver cells. Do the authors conclude that their animal studies are different than cell culture experiments with respect to liver response to hypoxia/ATP depletion? This figure should be omitted unless more information is provided.

To reflect the reviewer’s observation that, while still low, the p-values are above the typical 0.05 significance threshold, we have amended the text within the manuscript to note that there was only a trend toward increased expression. Given that the haemoglobin expression did not reach significance in the Atractyloside treated HEPG2 liver cells (which is concurrent with the hypoxia liver tissue samples), we have added an additional comment in the discussion that suggests further investigation in a skeletal muscle cell line might also reflect the trend we report in the hypoxic tissue samples.

  1. I am unclear about what Figure 8 is testing. The connection between HbB pulling down unknown ATPases and haem hypoxia control needs to be clarified.

The intention behind the experiments represented by figure 8 was to further reinforce the data presented in the earlier co-IP figures, and the in silico investigations. However, we appreciate the reviewer’s comment that this may pull down other unknown ATPase enzymes (not just F-type mitochondrial ATPase). We have therefore made the decision to remove this figure and any reference to it from the manuscript.

Minor comments

  1. It may be helpful to utilize a different mitochondrial loading control (e.g., Tom20) as although ß-actin and GAPDH can be found in the mitochondria, they are also found in the cytoplasm and may piggyback into the crude mitochondrial preps. This may help reduce the variability seen in some of the experiments.

We thank the reviewer for this suggestion as a means of reducing variability seen across samples within blots. Choice of control is always rather complicated, there is something to be said for a control that is informative for whole homogenates and mitochondrial preparations. But justifications can definitely be made both ways.

  1. Although “previously described” and referenced, it would be helpful if a brief description of the hypoxic protocols for the animal studies is presented.

A description of the hypoxic protocols for the animal studies has now been added to the manuscript.

Reviewer 2

The paper of Ebanks and collaborators aims at investigating the interaction between haemoglobin and mitochondrial proteins by using co-immunoprecipitation, in silico and in vitro experiments across multiple species. The paper is interesting and well written especially regarding the in silico analysis. In my opinion representative western blotting images of the graphs of figures 5, 6 and 7 should be shown in the figures. Moreover, according to the statistics, the expression of haemoglobin alpha and beta subunits in atractyloside treated HEPG2 cells does not seem to be increased, likely because of the small number of samples. I suggest that the authors rephrase the sentence or increase the number of the samples to confirm whether the statistically significant difference exists.

We thank the reviewer for taking the time to consider our manuscript and for the positive feedback that they have provided.

In response to their specific comments about the manuscript:

We had included western blot images that correspond to the graphs within figures 5, 6, and 7. However, feedback from collaborators concluded that inclusion of the blots made the necessarily smaller graphs difficult to see. We therefore chose to present them within the supplementary figures, so that the data can be viewed clearly.

As was highlighted by another reviewer, the increase in haemoglobin expression I HEPG2 does not reach a P value below the 0.05 threshold. We have therefore amended the manuscript to reflect that this is only a trend toward increased expression.

Review X (comments received after deadline)

Thank-you for your review and positive comments.

Minor comments

  1. Line 243 -> A large blank space is present in the manuscript with no
    real utility. Simple correction needed.

This will be removed in the editorial stage by the journal.

  1. The graphs are useful but need a bit of coherence, the templates are
    different, the p values are sometimes in plain text (figure, numbers) 
    and sometimes asterisks (*), just choose which you consider the best, I 
    may suggest numbers.

We have chosen to use asterisks where the p value has reached significance, and the decision to use numbers in figure 7 is to demonstrate the trend toward significance in the results of that experiment, but that the 0.05 threshold had not been crossed. We then chose to leave other, non-significant results without a number, to leave the graphs uncluttered.

  1. It may be useful, not only to discuss hypoxia but also “relative
    hypoxia” as it has already been suggested. I would be very interested in 
    the results in the future. (see refs.)

The concept of ‘relative hypoxia’ and the physiological response to this is an exciting area of research, and we too would be interested to see if the mitochondrial haemoglobin expression response is repeated in response to relative hypoxia of skeletal muscle.

Round 3

Reviewer 2 Report

If hemaglobin is soluble, then describing it "As previous studies have shown haemoglobin to localise to the inner mitochondrial membrane" the  authors should  clarify what they mean by "localize".

If " that haemoglobin was found to co-immunoprecipitate with ATP synthase and associated proteins, across a variety of different species, also with soluble, non-membrane protein ATPIF1." is true, then these references should be added to the paper. 

Author Response

Haemoglobin manuscript peer review round 2 – minor revisions

We thank the reviewer for taking the time to consider and then respond to our first round of manuscript revisions. The manuscript is significantly improved by those first comments, and these two final minor comments are addressed below.

  1. If haemoglobin is soluble, then describing it "As previous studies have shown haemoglobin to localise to the inner mitochondrial membrane" the authors should clarify what they mean by "localize".

To better clarify the meaning of “localise”, we have amended the statement referenced by the reviewer to say, “As a previous immunogold labelling transmission electron microscopy study has shown haemoglobin to be present in proximity to the inner mitochondrial membrane and its presence was detected by western blot of the inner mitochondrial membrane subfraction”. This work is from the manuscript published by Shephard et al. in 2014.

  1. If " that haemoglobin was found to co-immunoprecipitate with ATP synthase and associated proteins, across a variety of different species, also with soluble, non-membrane protein ATPIF1." is true, then these references should be added to the paper.

We apologise for the confusion generated by this statement in our first response to their comments; however, it refers to the data that we have presented within this manuscript, not to other manuscripts already published. However, our data is supported by published literature, including one study (which we cite within this manuscript) that demonstrated the co-immunoprecipitation of haemoglobin with ATP synthase subunits, the ADP/ATP translocase, and the mitochondrial phosphate ion exchanger (Brown et al., 2016). Further, a study by Brunyanszki et al. (2015) demonstrated the co-localisation of haemoglobin with mitochondrial complex I, a key component of the electron transfer system and directly upstream of the oxidative phosphorylation machinery.